# Gastric Carcinogenesis and Potential Role of the Transient Receptor Potential Vanilloid 1 (TRPV1) Receptor: An Observational Histopathological Study

**DOI:** 10.3390/ijms25158294

**Published:** 2024-07-30

**Authors:** Sylvester R. Groen, Daniel Keszthelyi, Arpad Szallasi, Jara A. van Veghel, Annick M. E. Alleleyn, Kata Csekő, Zsuzsanna Helyes, Iryna Samarska, Heike I. Grabsch, Ad A. M. Masclee, Zsa Zsa R. M. Weerts

**Affiliations:** 1Department of Gastroenterology and Hepatology, NUTRIM School of Nutrition and Translational Research in Metabolism, Maastricht University Medical Center+, 6629 HX Maastricht, The Netherlands; sylvester.groen@mumc.nl (S.R.G.); jara.van.veghel@catharinaziekenhuis.nl (J.A.v.V.); annickalleleyn@gmail.com (A.M.E.A.); z.weerts@mumc.nl (Z.Z.R.M.W.); 2Department of Pathology and Experimental Cancer Research, Semmelweis University, 1083 Budapest, Hungary; szallasi.arpad@med.semmelweis-univ.hu; 3Department of Pharmacology and Pharmacotherapy, Medical School, University of Pécs, 7624 Pécs, Hungary; csekoe.kata@gmail.com (K.C.); helyes.zsuzsanna@pte.hu (Z.H.); 4HUN-REN Chronic Pain Research Group, University of Pécs, 7624 Pécs, Hungary; 5National Laboratory for Drug Research and Development, 1117 Budapest, Hungary; 6Department of Pathology, Maastricht University Medical Center+, 6629 HX Maastricht, The Netherlands; i.samarska@mumc.nl (I.S.); h.grabsch@maastrichtuniversity.nl (H.I.G.); 7Division of Pathology and Data Analytics, Leeds Institute of Medical Research at St James’s University, University of Leeds, Leeds LS2 9JT, UK

**Keywords:** TRPV1, transient receptor potential vanilloid-1, gastritis, *H. pylori*, gastric cancer, intestinal metaplasia

## Abstract

The potential role of the transient receptor potential Vanilloid 1 (TRPV1) non-selective cation channel in gastric carcinogenesis remains unclear. The main objective of this study was to evaluate TRPV1 expression in gastric cancer (GC) and precursor lesions compared with controls. Patient inclusion was based on a retrospective review of pathology records. Patients were subdivided into five groups: *Helicobacter pylori* (*H. pylori*)-associated gastritis with gastric intestinal metaplasia (GIM) (n = 12), chronic atrophic gastritis (CAG) with GIM (n = 13), *H. pylori*-associated gastritis without GIM (n = 19), GC (n = 6) and controls (n = 5). TRPV1 expression was determined with immunohistochemistry and was significantly higher in patients with *H. pylori*-associated gastritis compared with controls (*p* = 0.002). TRPV1 expression was even higher in the presence of GIM compared with patients without GIM and controls (*p* < 0.001). There was a complete loss of TRPV1 expression in patients with GC. TRPV1 expression seems to contribute to gastric-mucosal inflammation and precursors of GC, which significantly increases in cancer precursor lesions but is completely lost in GC. These findings suggest TRPV1 expression to be a potential marker for precancerous conditions and a target for individualized treatment. Longitudinal studies are necessary to further address the role of TRPV1 in gastric carcinogenesis.

## 1. Introduction

Gastric cancer (GC) is the fifth most common cancer worldwide and is associated with significant cancer-related mortality [1,2]. Once GC has metastasized, the patient’s prognosis is poor. However, early diagnosis and treatment significantly increase patient survival [3]. Detection of early-stage GC bares difficulties due to a complex interaction of pathophysiological pathways and endo- and exogenous factors [4,5]. Chronic inflammation of the gastric mucosa can lead to the development of chronic atrophic gastritis (CAG), characterized by loss of specialized gastric glands, mucosal fibrosis and gastric intestinal metaplasia (GIM), a recognized precancerous lesion [6,7]. A major contributing factor is inflammation caused by *Helicobacter pylori* (*H. pylori*), especially when left untreated. In large epidemiologic studies, up to 3.0% of patients suffering from *H. pylori*-associated gastritis developed gastric carcinoma [8,9,10].

The potential role of dietary capsaicin (the bioactive pungent component of chili peppers) in GC development has long been subject to scientific debate [11,12,13], with no convincing evidence linking habitual capsaicin consumption to GC. However, some studies have suggested that chronic exposure or single high-dose capsaicin (30–60 milligrams per day) can induce gastric mucosal injury and (chronic) inflammation [14,15,16], whereas other studies suggested that capsaicin (particularly in lower doses corresponding to daily use) and its natural homologs and analogs (capsaicinoids) can prevent gastric mucosal damage related to alcohol and non-steroidal anti-inflammatory drugs and, therefore, may prevent gastric carcinogenesis [17,18].

Given the potential role of capsaicin in gastric carcinogenesis, specific interest has been directed in recent studies toward its target, the transient receptor potential Vanilloid 1 (TRPV1) receptor [19,20,21], although non-TRPV1-mediated effects of capsaicin have also been described [22]. The TRPV1 receptor is a calcium-permeable non-selective cation channel belonging to the transient receptor potential (TRP) cation channel family and is predominantly localized in extrinsic sensory fibers of the gastrointestinal tract and non-neuronal cells such as gastric epithelial cells [23,24]. TRPV1 receptor activation leads to the alterations of intracellular calcium concentrations, inducing the release of several neuropeptides from sensory nerve endings, exerting vascular smooth muscle action, inflammatory cell proliferation and plasma protein extravasation, which suppress gastric acid production and may prevent apoptosis and oxidative stress [25,26].

Regarding carcinogenesis, the involvement of TRPV1 as a tumor suppressor has been described in various carcinogenic pathways, including breast cancer, urothelial cell carcinoma and thyroid cancer [27,28]. However, the upregulation of TRPV1 in colonic cancer and esophageal squamous cancer correlates with tumor progression, migration, and poor survival [29,30]. In contrast, in vitro (over)expression of TRPV1 in mouse models of GC resulted in the inhibition of tumor proliferation, migration, tissue invasion and smaller tumor size. Others have found less peritoneal dissemination in TRPV1-depleted mice [31,32]. In human GC cells, a significant decrease in TRPV1 expression was described, suggesting TRPV1 to be a potential marker for the prognosis of GC due to its correlation with GC progression [33,34]. Considering the current literature, the role of TRPV1 expression in gastric carcinogenesis remains unclear. The main objective of this observational study was to assess TRPV1 expression in the gastric mucosa in patients with *H. pylori*-associated gastritis with and without GIM, CAG and intestinal-type adenocarcinoma.

## 2. Results

### 2.1. Study Population and Samples

Fifty-five patients were included for analysis, of whom 31 (65.4%) were females. The median age was 60.4 (±16.3) years, and 58.2% had an age over 60 years. Patients were distributed into five groups: five patients were labeled as controls with normal stomach mucosa, 12 patients with *H. pylori*-associated gastritis with GIM, 13 patients with CAG with GIM, 19 patients with *H. pylori*-associated gastritis without GIM and six patients with gastric adenocarcinoma. In all patients, at least one stomach tissue sample was retrieved during biopsy for analysis and immunohistochemical staining. In most patients, gastric biopsies were retrieved from the antral and body region of the stomach (n = 52, 94.5% of total). In three patients, only cardia (n = 2) or angulus (n = 1) were sampled. All patient characteristics are provided in Table 1.

### 2.2. TRPV1 Expression Is Lost in Gastric Adenocarcinoma

After initial pathological examination, tissue samples underwent immunohistochemical staining for the assessment of TRPV1 immunopositivity (Figure 1A–D). In all tissue samples, the median immunoreactive score (IRS, paragraph 4.3 of the materials and methods section) was calculated to quantify TRPV1 expression, as outlined in Table 2. Additionally, a median IRS was calculated for each group of all cells combined (foveolar cells, parietal cells and chief cells). In parietal cells, the TRPV1 expression was generally higher in all groups. In foveolar cells of gastric tissue retrieved from controls, there was no TRPV1 expression; therefore, no median IRS could be calculated. In tissue samples identified as gastric adenocarcinoma, there was no TRPV1 expression in either foveolar cells, parietal cells or chief cells.

### 2.3. TRPV1 Expression Significantly Higher in GIM Compared with No GIM or Controls

Between all groups, there were significant differences in TRPV1 expression, in which a significantly greater TRPV1-immunopositivity was found in patients with GIM compared with patients without GIM and controls (Table 2). Similar results were seen when all cells were combined to calculate a mean IRS for each group (Figure 2A). This significant difference was confirmed using post hoc analysis, except for the inter-group difference of *H. pylori* versus GIM and CAG versus GIM (*p* = 0.13).

### 2.4. Significant Differences TRPV1 Expression within All Groups according to Cell Type

In controls, TRPV1 expression was significantly higher in parietal cells, followed by the TRPV1 expression in chief cells, and was the lowest in foveolar cells (F = 92.250, *p* < 0.002). Similar results were conducted within other groups, all statistically significant: *H. pylori*-associated gastritis with GIM (F = 12.123, *p* = 0.002), CAG with GIM (F = 10.583, *p* = 0.003), and *H. pylori*-associated gastritis without GIM (F = 9.534, *p* = 0.002). Post hoc analysis revealed the highest TRPV1 expression in parietal cells and the lowest in foveolar cells in all groups.

## 3. Discussion

In this observational histopathological study, TRPV1 expression was significantly higher in patients with *H. pylori*-associated gastritis compared with controls. When GIM was present, TRPV1 expression was even higher compared with patients without GIM, particularly in parietal cells. TRPV1 expression was completely absent in gastric adenocarcinoma.

The role of TRPV1 in various inflammatory and carcinogenic pathways has been previously studied with contradictory results, revealing both pro- and anti-inflammatory and tumor-promoting and -suppressing effects. Several studies have demonstrated an increase in TRPV1 expression in various inflammatory processes, in which downregulating the TRPV1 receptor by genetic manipulation or inhibiting with an antagonist contributed to alterations in disease activity and severity [1,4,35]. This effect was shown in animal studies addressing inflammatory bowel disease, demonstrating decreased inflammation and disease severity in TRPV1-depleted mice [24,36,37]. In the gastric mucosa, the tachykinin substance P and CGRP are released in response to TRPV1 activation, which triggers neurogenic inflammatory pathways with consecutive inflammatory cell activation [19,27,35]. *H. pylori*-induced gastric inflammation is associated with the upregulation of pro-inflammatory cytokines such as IL-6 and IFN-Ƴ, pathways in which the TRPV1 receptor activation can have a facilitating role [8,9,38]. On the other hand, other studies have shown that a nontoxic dose of capsaicin inhibited the *H. pylori*-induced IL-8 production by gastric epithelial cells through the modulation of I-κB-, NF-κB- and IL-8 pathways [15].

*H. pylori*-associated gastritis is known as an individual risk factor for the development of GC, especially when untreated [8]. The co-administration of high-dose capsaicin (30–60 milligrams/day) has shown a synergistic contribution to accelerated loss of differentiated gastric epithelial cell types, leading to precancerous conditions such as GIM and CAG [8,9,15,18]. The significant increase in TRPV1 expression in patients with *H. pylori*-associated gastritis compared with controls might be due to the pro-inflammatory roles of TRPV1. A pending knowledge gap remains: it is unclear whether *H. pylori*-associated gastritis leads to TRPV1 overexpression, thereby increasing the risk of gastric cancer, or if the secondary epithelial changes associated with TRPV1 expression result in (pre)malignant transformation. In addition, it remains to be established whether TRPV1-induced immune activation contributes to the further aggravation of *H. pylori*-gastritis or rather has mitigating effects. Future preclinical functional and prospective clinical research investigating the specific correlations of TRPV1 expression in *H. pylori*-associated gastritis and gastric carcinogenesis is needed.

A complete loss of TRPV1 expression in gastric adenocarcinoma was found in our study population. This is largely in line with findings as described by Gao et al., in which a decreased TRPV1 expression in human GC was seen compared with tumor-adjacent tissue [4]. This decrease was associated with tumor size, metastases and poor prognosis, suggesting a tumor suppressant role of TRPV1. The complete loss of TRPV1 in gastric carcinoma, as observed here, may imply a tumor suppressor role. Our study also showed a significantly higher TRPV1 expression in patients with GIM and CAG, both known as precancerous conditions [1,6]. The number of reports underlining the involvement of calcium-permeable TRP channels in GC development is increasing, showing either aberrant receptor expression or function leading to alterations in intracellular calcium levels [4,33,39]. Over the past two decades, several members of the TRP channel family besides TRPV1 have been identified in gastric carcinogenesis, including TRPV2/4/6, TRP Melastatin (TRPM) 2/5/7/8 and TRP Canonical (TRPC) 1/3/6 [25,40,41,42,43,44]. However, most of these receptors are associated with upregulation or overexpression in gastric cancer and have been suggested as oncogene and tumor promotors in GC. Furthermore, the expression of TRPV2, TRPV4 and TRPV6 was found to increase according to the tumor stage with an association with poor prognosis, particularly in advanced stages [4,40]. Additionally, the inhibition of various TRP channels, including TRPM2/8 and TRPV2/4, may suppress the progress of GC development. With regard to TRPV1, its activation leads to intracellular calcium increase and the activation of anti-oncogenic pathways, resulting in decreased expression of cyclin D1 and MMP2, among others. In line with this concept, the downregulation of TRPV1 expression in mice promoted GC cell proliferation, invasion and dissemination, suggesting a tumor suppressant role of TRPV1 [4]. In contrast, another study found that the upregulation of TRPV1 expression after the administration of a high dose of (dietary) capsaicin (30–60 milligrams/day) can accelerate GC metastasis [1]. Since there is significantly higher TRPV1 expression in patients with GIM or CAG compared with controls, the question arises whether this is due to a tumor-promoting effect eventually resulting in GC or a counterbalancing tumor-suppressant mechanism directed at mitigating the pro-cancerous processes. The eventual loss of TRPV1 expression is supportive of the latter notion. Hypothetically, TRPV1 expression could be considered a marker for precancerous conditions. Future longitudinal studies with serial assessment of TRPV1 staining as a part of surveillance endoscopies for gastric precancerous conditions could further clarify this issue. Moreover, it warrants further investigation to determine whether there is a relationship between capsaicin dose and its potential pro- or anti-carcinogenic effects and whether these effects are mediated by TRPV1-dependent mechanisms.

Due to the retrospective nature of this observational immunopathological study with a small sample size, there are several limitations to be noted. Since this study had a purely observational nature, no longitudinal patient follow-up was performed. Therefore, it remains uncertain whether patients with precursor stages of gastric cancer did or did not develop malignant neoplasms. Additional verification of the staining of the immonohistochemistry of the pathological preparations would clarify this further. In addition, no functional readouts were conducted. Although errors in the immunohistochemical staining process could have occurred, mostly leading to underestimating TRPV1 immunopositivity, a significant increase in TRPV1 expression was found. This effect could, therefore, be even more apparent when correcting for this bias.

In summary, this observational histopathological study is the first to address TRPV1 expression in precursor stages of GC and *H. pylori*-associated gastritis, which is an individual risk for GC development. Significantly increased TRPV1 expression was demonstrated in *H. pylori*-associated gastritis and was even greater in patients with GIM and CAG, but its complete loss was seen in GC. These findings point to a potential tumor-suppressive role of TRPV1 and suggest that TRPV1 expression is a potential biomarker for precancerous conditions and a putative novel target for individualized treatment. Future longitudinal clinical studies with follow-up of precursor stages of GC and those addressing the impact of TRPV1 agonists (e.g., capsaicin) should provide more insight into the exact role of TRPV1 in gastric carcinogenesis.

## 4. Materials and Methods

### 4.1. Study Design and Patient Selection

In this observational study, samples were selected based on a retrospective review of the pathology records of patients who underwent upper gastrointestinal (GI) endoscopy at Maastricht University Medical Center+ (MUMC+) in Maastricht, the Netherlands, between 2009 and 2017. Patient inclusion was based on a histopathological diagnosis, which was confirmed independently by two pathologists. Patients were subdivided into five groups based on pathologic diagnosis: *H. pylori*-associated gastritis without GIM, *H. pylori* infection-associated gastritis with GIM, CAG, adenocarcinoma (intestinal type carcinoma) and controls. In all patients, multiple biopsies were taken from the stomach according to the Sydney protocol, using standard biopsy forceps with a diameter of 2.8 mm (Boston Scientific, Marlborough, MA, USA). The initial indication for performing an upper GI endoscopy included upper GI symptoms (p.e. regurgitation, upper stomach pain), iron deficiency and anemia. Controls were selected from the pathological records of patients who underwent upper GI endoscopy to exclude celiac disease, in which multiple biopsies were taken in the stomach and duodenum. After excluding celiac disease and other gastric pathological conditions using histopathological evaluation as described earlier, patients were assigned to the control group.

### 4.2. Ethical Statement and Tissue Sampling

The study was approved by the Local Ethic Committee (METC number 16-4-012) of the Maastricht University Medical Center+ (MUMC+). After initial histopathological examination in MUMC+, human tissue samples were anonymized for secondary immunohistochemical assessment and securely shipped to the laboratory of the University of Pécs/Szentágothai Research Center in Pécs, Hungary. Tissue samples were digitalized after immunohistochemical staining in an anonymized electronic database.

### 4.3. TRPV1 Immunohistochemical Staining

Tissue samples were formalin-fixed and paraffin-embedded, and 5 µm sections were cut as part of a routine pathological specimen work-up. For secondary analysis of TRPV1 expression, tissue samples were deparaffinized, rehydrated and incubated in an acidic citrate buffer (pH = 6) in a microwave oven for antigen recovery. Endogenous peroxidase activity was quenched using 3% hydrogen peroxide. After sections were washed and incubated in a blocking solution, anti-TRPV1 antibodies were applied (GP14100; Neuromics, Edina, MN, USA). Slides were then incubated with an anti-rabbit antibody conjugated with the EnVision system with horse radish peroxidase (DakoCytomation, Carpinteria, CA, USA). To visualize the reaction, 3,3-diaminobenzide tetrachloride was used, followed by counterstaining with hematoxylin [45]. Validation of antibody selectivity was validated by the lack of immunopositivity after the application of the blocking peptide (Neuromics, Edina, MN, USA) based on previous studies [46].

### 4.4. Quantification of TRPV1 Expression

The quantitative assessment of TRPV1 immunopositivity was performed in every tissue sample in the three main stomach cells: foveolar cells, parietal cells and chief cells. This quantitative assessment was based on the intensity of the immunohistochemical staining and the proportion of immunopositive cells on 10 fields of vision/slide/biopsy by an expert pathologist (A.S.) blinded to the clinical results of the initial pathology analysis (performed by H.G. and I.S.). Only the results of the quantitative assessment are reported in this study. Immunohistochemical staining density was scored on a four-point scale: no immunoreactivity (0), weak staining (1), moderate staining (2) and strong staining (3) (Table 3, A). The proportion of immunopositive cells was scored as none (0), less than 25% of the area (1), 25–49% of the area (2), 50–74% of the area (3) and more than 75% of the area (4) (Table 3, B). To summarize the quantification of TRPV1 expression in biopsies, the IRS was calculated as follows:Intensity scale (A) X proportion of positive staining (B).Scores were subsequently grouped to form the IRS (scale 1–12).


In addition, the IRS provides quantitative TRPV1 expression on a four-point scale: negative (0–1), slightly positive (2–3), moderately positive (4–8) and strongly positive (9–12) (Table 3, C) [47]. Tissue samples were digitalized for evaluation using a Pannoramic Digital Slide Scanner with CaseViewer software version 2.4 (3D HISTECH Ltd., Budapest, Hungary).

### 4.5. Data and Statistical Analysis

Statistical analysis was performed using SPSS statistics 28.0 (IBM, Armonk, NY, USA) and R version 4.4.0 (R Core Team, 2024. R: A language and environment for statistical computing. Vienna, Austria). Continuous data were presented as medians and interquartile ranges (IQRs) and categorical data as proportions (%). To compare (non-parametric) continuous data between subgroups, the Kruskal–Wallis test and post hoc Tukey’s test were conducted. To compare (non-parametric) continuous data within subgroups, a repeated measures ANOVA was conducted, including Mauchly’s Test of Sphericity, to assess the assumption of sphericity. Greenhouse–Geisser corrections were applied where necessary. Post hoc comparisons were conducted by using a paired sample t-test with Bonferroni correction for multiple testing. A P value of <0.05 was deemed statistically significant. Considering the exploratory nature of the study, no formal sample size calculation was performed.

## Figures and Tables

**Figure 1 ijms-25-08294-f001:**
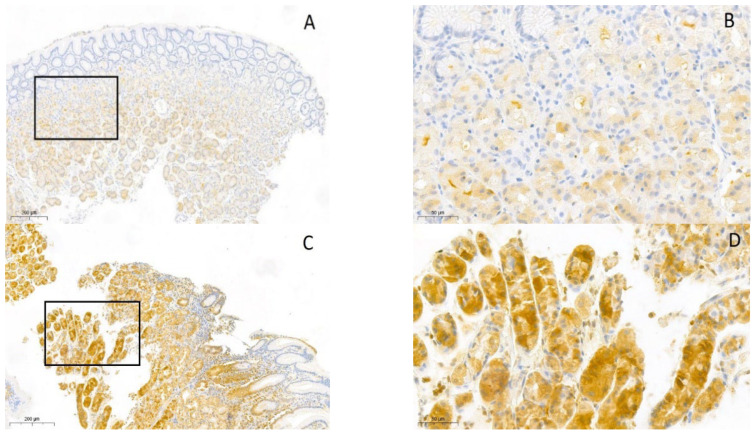
Microscopic pictures of gastric biopsies after immunohistochemistry. Immunohistochemical staining (TRPV1 expression) is in brown, and hematoxylin counterstain is in blue. (**A**,**B**) TRPV1 expression in gastric antral type mucosa controls (10× and 40×). (**C**,**D**) TRPV1 expression in *H. pylori*-associated gastritis in gastric antral type mucosa (10× and 40×). (**E**,**F**) Complete loss of TRPV1 expression in gastric adenocarcinoma antral type mucosa (10× and 40×).

**Figure 2 ijms-25-08294-f002:**
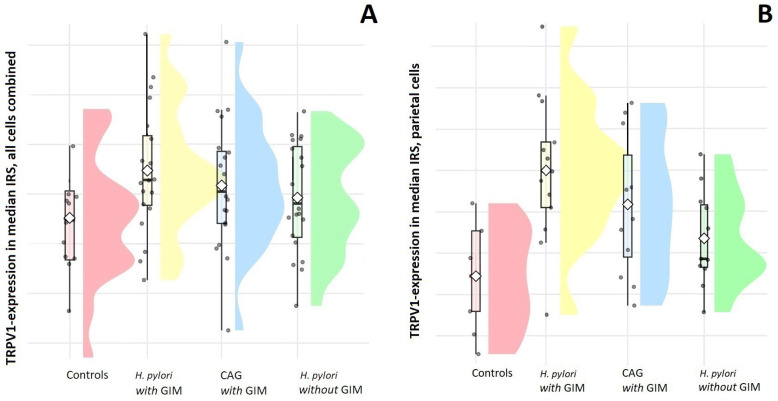
Raincloud plot showing the distribution of TRPV1 expression as quantified in a median IRS: controls, *H. pylori*-associated gastritis with gastric intestinal metaplasia (*H. pylori* with GIM), chronic atrophic gastritis with GIM (CAG with GIM) and *H. pylori*-associated gastritis without GIM (*H. pylori* without GIM). The plot provides a detailed view of the distribution, central tendency and variability of TRPV1 expression (in IRS) within each group. (**A**) Raincloud plot of the median IRS all cells combined—showing a significantly higher TRPV1 expression in patients with GIM compared with those without GIM and controls. (**B**) Raincloud plot showing a significantly higher TRPV1 expression in patients with GIM compared with those without GIM and controls in parietal cells.

**Table 1 ijms-25-08294-t001:** Baseline characteristics.

Characteristic		Controls (n = 5)	*H. pylori* with GIM (n = 12)	CAG with GIM (n = 13)	*H. pylori*, without GIM (n = 19)	Gastric Adeno- Carcinoma (n = 6)
Age, median, years (±SD)		49.60 (11.70)	65.75 (10.49)	57.85 (14.18)	55.95 (19.25)	78.33 (6.71)
	<60 years, n (%)	3 (60.0)	3 (25.0)	7 (53.8)	10 (52.6)	N.A.
	≥60 years, n (%)	2 (40.0)	9 (75.0)	6 (46.2)	9 (47.4)	6 (100)
Gender						
	Male, n (%)	1 (20.0)	6 (50.0)	3 (23.1)	11 (48.9)	3 (50.0)
	Female, n (%)	4 (80.0)	6 (50.0%)	10 (76.9)	8 (42.1)	3 (50.0)
Biopsy location stomach						
	Antrum only, n (%)	3 (60.0)	2 (16.7)	N.A.	N.A.	N.A.
	Antrum and body, n (%)	2 (40.0)	10 (83.3)	13 (100)	19 (100)	5 (83.3)
	Other regions *, n (%)	N.A.	N.A.	N.A.	N.A.	1 (16.7)

* including biopsies of cardia and angulus. Abbreviations: *H. pylori* = *H. pylori*-associated gastritis; GIM = gastric intestinal metaplasia; CAG = chronic atrophic gastritis; GC = gastric carcinoma; N.A. = not applicable.

**Table 2 ijms-25-08294-t002:** TRPV1 expression in median IRS.

						Kruskal–Wallis Test *	*p*-Value **
	Controls (n = 5)	*H. pylori* with GIM (n = 12)	CAG with GIM n (n = 13)	*H. pylori* without GIM (n = 19)	Gastric adeno-carcinoma (n = 6)		
**TRPV1 expression in** **Median IRS (±SD)**							
Foveolar cells	N.A.	6.00 (3.59)	5.23 (2.62)	3.26 (3.08)	N.A.	31.10	**<0.001**
Parietal cells	3.60 (0.54)	10.25 (2.01)	8.77 (1.79)	6.58 (2.77)	N.A.	34.07	**<0.001**
Chief cells	5.60 (2.190	6.50 (2.81)	6.00 (2.45)	5.84 (2.14)	N.A.	17.68	**0.001**
Mean IRS, all cells	3.07 (0.86)	7.58 (1.23)	6.67 (1.62)	5.23 (1.88)	N.A.	34.73	**<0.001**
***p*-Value intra-group** **comparison ****	**0.002**	**0.002**	**0.002**	**0.003**	N.A.		

* The test statistics are adjusted for ties. ** Post hoc analysis is described in the text. Abbreviations: *H. pylori* = Helicobacter pylori-associated gastritis; GIM NEG = without gastric intestinal metaplasia; GIM POS = with gastric intestinal metaplasia; CAG = chronic atrophic gastritis; IRS = immunoreactive score; SD = standard deviation; N.A. = not applicable.

**Table 3 ijms-25-08294-t003:** Explanation and interpretation of the immunoreactive score (IRS).

A (Intensity Scale)	B (Proportion of Positive Staining in %)	IRS Score * (Multiplication A × B)
0 = no immunoreactivity	0 = no stained area	0–1 = negative
1 = weak staining	1 = <25% of stained area	2–3 = slightly immunopositive
2 = moderate staining	2 = 25–49% stained area	4–8 = moderate immunopositive
3 = strong staining	3 = 50–74% stained area	9–12 = strongly immunopositive
	4 = >75% stained area	

* Final IRS score (A × B) = 0–12.

## Data Availability

Data is contained within the article.

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
