# Peer review of "Gastric Carcinogenesis and Potential Role of the Transient Receptor Potential Vanilloid 1 (TRPV1) Receptor: An Observational Histopathological Study"

_ijms, 2024, doi:10.3390/ijms25158294_

Round 1

Reviewer 1 Report

Comments and Suggestions for Authors

Retrospective observational study, based on the review of the medical records of patients with gastric pathology who were divided into five groups. The aim of the study was to evaluate the expression of TRPV1 in the gastric mucosa in patients with H. pylori-associated gastritis with and without gastric intestinal metaplasia, chronic atrophic gastritis and in intestinal-type adenocarcinoma. Beyond the results obtained, I immediately draw attention to two quite important points of weakness such as: the lack of longitudinal follow-up of the patients and the second, the lack of verification of the staining of the immonohistochemistry of the pathological preparations. The work is based on the Transient Receptor Potential Vanilloid 1 receptor and its importance in the neoplastic genesis of stomach cancer and its subsequent possibility of metastasis. This paper reiterates a concept that is already part of the scientific community's baggage but which is important to read given that it constitutes a further brick for the construction of the temple of understanding all the chemical and biochemical mechanisms that lead to the modification of the genes that will subsequently generate the neoplasm. Good English, excellent iconography, the bibliography, especially with 5-6 articles, supports the theses of colleagues

Comments on the Quality of English Language

english needs to be revised

Reviewer 2 Report

Comments and Suggestions for Authors

1.I am not sure if this paper matches the standards of this journal. It makes me wonder if the journal's quality has declined.

2.There are other papers suggesting that TRPV2, M6, M7, M8, and M2 are related to gastric cancer. It is necessary to confirm their relevance in human tissues. Additionally, I would like these to be discussed in the discussion section.

3.This paper is significant in confirming the relevance of TRPV1 in human gastric cancer, but it has limited data and lacks experiments on the relevance of other TRP ion channels. Therefore, I do not consider the quality of the paper to be high.

Round 2

Reviewer 2 Report

Comments and Suggestions for Authors

It is well revised.

Author Response

Dear reviewer. We would like to thank you again for your constructive feedback on our manuscript “Gastric carcinogenesis and potential role for the transient receptor potential vanilloid 1 (TRPV1) receptor: an observational histopathological study”. We have carefully considered all comments and suggestions for revision and made changes in the final version of our manuscript.